# Beyond Static Endpoints: Tool Programs as an Interface for Flexible Agentic Web Services

**Mugeng Liu**[1]  **Shuoqi Li**[2]  **Yixuan Zhang**[1]  **Yun Ma**[3]

## Abstract

In the agentic web era, LLM-based agents increasingly invoke web services as tools, yet most interfaces remain *static endpoints* that poorly express long-horizon workflows with loops, conditionals, joins, and retries. We present TOOLPRO, which represents an agent's tool intent as an *executable tool program* that compactly encodes multi-step service interactions with explicit effect types. TOOLPRO combines constraint-guided program construction, effect-aware replay for exactly-once state-modifying calls, and a profile-driven policy that decides when program execution outperforms stepwise calling. We instantiate TOOLPRO over MCP-style services with WebAssembly sandboxing and evaluate it on diverse workflows of real-world applications. TOOLPRO reduces end-to-end latency by up to 53.4% and client-side traffic by up to 96.1%, with larger gains under higher network latency and workflow complexity.

## 1. Introduction

LLM-based agents (Yao et al., 2023; Schick et al., 2023; Liu et al., 2024; Qin et al., 2024; Liu et al., 2026) are increasingly expected to complete long-horizon workflows by orchestrating web services. Yet most services are still exposed through *static API endpoints*—an interface designed for single-shot queries, not for procedural, multi-step interaction. When a task requires control flow (e.g., loops, conditionals), intermediate bindings, or intent-dependent data access, an agent must externalize the workflow into a brittle sequence of endpoint calls interleaved with multi-round reasoning (Yao et al., 2022; Deng et al., 2023; Zhou

[1]School of Computer Science, Peking University, Beijing, China [2]School of Software & Microelectronics, Peking University, Beijing, China [3]Institute for Artificial Intelligence, Peking University, Beijing, China. Correspondence to: Yun Ma <mayun@pku.edu.cn>.

*Proceedings of the 43rd International Conference on Machine Learning*, Seoul, South Korea. PMLR 306, 2026. Copyright 2026 by the author(s).

et al., 2024). The stepwise interface scales poorly, multiplying network turns, systematically over- and under-fetching data, and triggering cascading retries with inconsistent side effects upon partial failure.

Our insight is that the inefficiency is fundamentally *representational*, as shown in Figure 1. Endpoint sequences are a weak interface for expressing tool intent, because they fragment a coherent multi-step plan into local decisions conditioned on intermediate responses. As a result, both client–service round trips and agent reasoning rounds grow with the number of procedural steps, while failures amplify as a single mismatch can cascade into retries and state inconsistencies. Agentic workflows need an interface that can express *"perform this multi-step interaction"* as a single, composable object whose execution can be delegated, optimized, and checked.

To this end, we propose *tool programs* as an executable representation of tool intent. These programs compactly encode a multi-step service interaction, complete with control flow and intermediate bindings. Furthermore, they feature explicit effect types that distinguish state-preserving READ operations from state-modifying WRITE operations. Rather than repeatedly selecting endpoints on the client, an agent synthesizes a tool program and delegates its execution to a controlled service-side runtime. Making intent first-class enables three capabilities that static endpoints do not provide, including (i) *turn reduction* by consolidating multi-step interactions into fewer network round trips, (ii) *effect-aware execution* by enforcing different semantics for READ versus WRITE, and (iii) *safe re-execution* under repair without duplicating side effects. Realizing tool programs in practice raises three challenges: (1) Executability. LLM-produced programs may fail to compile or crash at runtime under a typed, sandboxed substrate. (2) Side effects under repair. Partial success before failure makes naive re-execution duplicate WRITEs and corrupt service state. (3) When to consolidate. For short workflows or low-latency networks, program construction overhead can outweigh the savings from turn reduction.

By addressing these challenges, we present TOOLPRO, the first agentic web service runtime that operationalizes **Tool Pro**grams as an interface for flexible agentic web services.

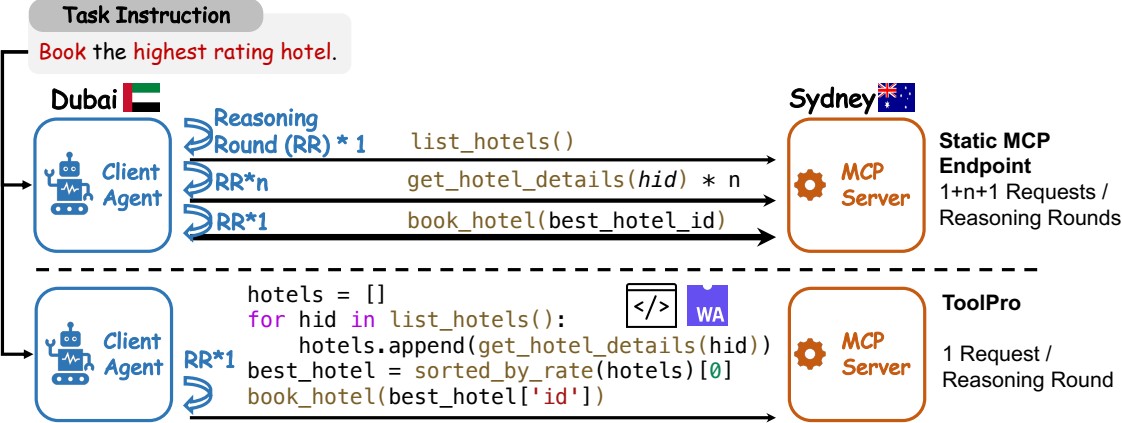

Figure 1. **From static endpoints to tool programs.** Stepwise endpoints force the agent to repeatedly call endpoints and re-prompt to realize control flow. Tool programs (TOOLPRO) instead package a multi-step interaction as one executable object with explicit effects, enabling service-side execution and safe re-execution under repair.

Specifically, to improve executability, TOOLPRO introduces constraint-guided program construction, which combines lightweight formatting constraints with compiler/runtime feedback, and resolves common failures via service-side repair to avoid repeated client–server back-and-forth. To control side effects, TOOLPRO enforces effect-aware replay that provides *exactly-once* semantics for WRITE operations across iterative repair and re-execution. To decide when consolidation is worthwhile, TOOLPRO applies a profile-driven consolidation rule that adaptively selects between stepwise calling and program execution.

We implement TOOLPRO over MCP-style web services using WebAssembly sandboxing and evaluate it on diverse workflows within realistic applications. TOOLPRO reduces end-to-end latency by up to 53.4% and client-side traffic by up to 96.1%, with gains increasing under higher network latency and workflow complexity. These results highlight that TOOLPRO lays an important foundation for agent-facing service interfaces in the emerging agentic web.

This paper makes the following contributions.[1]

- We identify static endpoints as a representational bottleneck for agentic web workflows and propose *tool programs* as an agent-facing service interface.
- We present TOOLPRO to make tool programs practical by addressing core challenges. It employs constraint-guided construction for executability, effect-aware replay with exactly-once semantics for safe repairs, and a profile-driven policy for adaptive consolidation.
- We evaluate TOOLPRO on real applications and workflows, demonstrating substantial reductions in latency and client-side traffic, highlighting a promising direction to build an efficient agentic web.

---

[1]Code is publicly available at `https://github.com/morgen52/toolpro_icml26`.

## 2. Problem Formulation

We study agentic web-service tool use where a client-side LLM agent orchestrates server-side web services to complete procedural workflows. A tool-facing service exposes a set of endpoints $\mathcal{E}$ over an internal service state $s$. A call is $\text{CALL}(e, a)$ for $e \in \mathcal{E}$ and arguments $a$, returning an output $o$ (or an error) and possibly updating $s$. We focus on procedural *agentic workflows* that inherently require control flow, intermediate bindings, and intent-dependent data access.

**Bottleneck: stepwise endpoint sequences.** With *static endpoints*, an agent realizes an intent via a stepwise interaction loop. At step $i$, it decides the next call $(e_i, a_i)$ conditioned on the task context and past observations $(o_1, \ldots, o_{i-1})$, then issues the request and observes $o_i$. This yields a call sequence $\pi = \langle (e_1, a_1), \ldots, (e_N, a_N) \rangle$ interleaving $N$ client–service round trips with $N$ client-side decision rounds. This interface makes the procedure *reactive*, which fragments a coherent procedure into client-side next-call decisions. As procedural length grows, this interface (i) inflates latency via repeated RTT and per-step decision overhead, (ii) induces over-/under-fetching because control logic must be implemented outside the service, and (iii) makes recovery brittle, with partial failures triggering retries that can duplicate state-modifying operations and corrupt state.

**Key idea: tool programs as an interface.** We propose to make tool intent first-class by representing a workflow as an *executable tool program*. A tool program $P$ is a procedure whose atomic operations are endpoint invocations $\text{CALL}(e, a)$, composed with structured control flow and intermediate bindings. Given an initial state $s$, executing $P$ produces a result $y$ and a (possibly updated) state $s'$. This shifts the interface from next-call selection to program submission, enabling service-side execution, optimization, and checking while preserving the service's observable behavior.

**Goals.** We aim to make tool programs practical *without* changing the observable outcomes of the underlying service interaction and *without* introducing prohibitive overhead compared to stepwise calling. Concretely, we target (G1) *effect-safe observable semantics* under failures and retries; and (G2) *improved end-to-end efficiency* in latency and client-side traffic.

*(G1)* Executing a tool program may fail (e.g., due to runtime errors in generated code), triggering repair and re-execution. We aim for an *interface-level* guarantee: re-execution for the same high-level intent must not introduce additional side effects beyond stepwise execution. Note that we do not attempt to eliminate endpoint-level failures, which are inherent to the underlying service and can equally occur under stepwise execution.

To reason about side effects, we assume each endpoint has an effect label $\mathrm{eff}(e) \in \{\mathrm{READ}, \mathrm{WRITE}\}$, distinguishing state-preserving queries from state-modifying operations. An execution induces a trace $\tau(P) = \langle(e_1, a_1, o_1), \ldots, (e_N, a_N, o_N)\rangle$, where only WRITE calls may change $s$. We target two interface-level properties: (i) *observational equivalence*: conditioned on the same underlying sequence of endpoint outcomes, program execution exposes the same outputs/errors as stepwise execution; and (ii) *retry safety*: across repair-driven re-executions for the same intent, WRITE effects are not duplicated, yielding interface-level *exactly-once* semantics.

*(G2)* Tool programs are beneficial only when their one-time construction cost is amortized by reducing round trips. A simple latency model contrasts stepwise calling and program execution, as follows:

$$T_{\mathrm{STEP}} \approx \sum_{i=1}^{N}\big(T_{\mathrm{RTT}} + T_{\mathrm{DEC}} + T_{\mathrm{API}}\big),$$

$$T_{\mathrm{PROG}} \approx T_{\mathrm{BUILD}} + \Big(T_{\mathrm{RTT}} + \sum_{i=1}^{N} T_{\mathrm{API}}\Big), \qquad (1)$$

where $T_{\mathrm{RTT}}$ is client–service RTT, $T_{\mathrm{DEC}}$ is per-step decision overhead on the client-side agent, $T_{\mathrm{API}}$ is endpoint execution time, and $T_{\mathrm{BUILD}}$ includes program construction, compilation, and potential repair. Client-side traffic follows a similar trade-off. Stepwise calling repeatedly transmits requests and prompts across $N$ rounds, whereas tool programs consolidate interaction into a small number of uploads and responses.

**Challenges.** This formulation surfaces three challenges in operationalizing tool programs in practice: (C1) *executability* of LLM-produced programs under a typed, sandboxed substrate; (C2) *exactly-once effect semantics* for WRITE calls under repair and re-execution; and (C3) *adaptive consolidation* to decide when program execution is more efficient than stepwise calling.

## 3. TOOLPRO Design

TOOLPRO makes *tool programs* a first-class service interface for agentic workflows. Given an intent instance and tool specifications, the agent submits a single effect-typed program $P$; the service-side runtime then compiles, optionally repairs, and executes $P$ as a unit while enforcing an interface contract. TOOLPRO achieves this interface shift by addressing the three challenges (introduced in §2) in operationalizing tool programs in practice.

TOOLPRO follows a *synthesize–project–compile–execute* pipeline with a conservative fallback path. (1) Given task intent and service endpoints, the client synthesizes a candidate tool program and performs lightweight structural checks to reject obviously misaligned programs early. (2) The server projects the program into a constrained interface-program surface that is analyzable and enforceable, then compiles and executes it in a sandbox. (3) If compilation or execution fails, the server performs bounded in-place repair using compiler diagnostics and runtime traces as verifiable feedback. During any execution and re-execution, the runtime mediates external calls and enforces effect-aware replay to prevent duplicated WRITE effects. Finally, a profile-driven consolidation policy decides whether to use program execution or revert to stepwise calling for the current task.

### 3.1. Tool Programs as an Interface

TOOLPRO employs the *tool program* as the submitted interface object and the contract that the runtime enforces for any accepted program.

The tool program serves as the interface object, which is the unit of interaction at the interface. A client represents an agentic workflow as a tool program $P$ (defined in §2) and submits it to the runtime for execution as a unit. The only way for $P$ to interact with the underlying service is via a unified stub $\mathrm{CALL}(e, a)$, where each dynamic call (a runtime instance of $\mathrm{CALL}$) triggers one endpoint invocation and returns an output (or error) observable to the program. This makes the workflow logic explicit and inspectable on the server side.

To make such a program $P$ enforceable and safe as an interface, the runtime cannot accept arbitrary code, which motivates a constrained surface. TOOLPRO restricts submitted programs to a constrained surface that expresses service-interaction logic rather than arbitrary application logic. Concretely, the runtime enforces: (i) structured control flow only (`if`/`else`, `for`/`while`); (ii) external interaction only through $\mathrm{CALL}(\cdot)$ with explicit effect annotations at each call site; (iii) no exceptions, threads, dynamic linking, or unsafe memory operations; and (iv) no ad-hoc networking or filesystem access beyond the tool-facing service boundary. These restrictions ensure that compilation errors, runtime

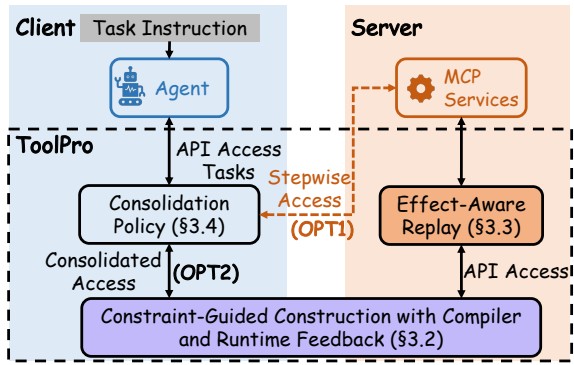

*Figure 2.* Three mechanisms of TOOLPRO.

traces, and call events refer to a stable, analyzable surface that the runtime can repair and mediate reliably.

Moreover, TOOLPRO makes side effects explicit via effect typing. Concretely, every external call site in $P$ must declare an effect label (READ/WRITE), which the runtime checks against the endpoint's declared effect $\text{eff}(e)$. These annotations serve as the handle for retry protection. READ calls may be safely re-issued, whereas completed WRITE calls must be replay-protected under repair-driven re-execution.

Building on the interface object, constrained surface, and effect-typed boundaries, TOOLPRO enforces the following contract for any well-formed program $P$.

- **Program order.** TOOLPRO processes dynamic calls in program order and never reorders external invocations.

- **Observable preservation.** Conditioned on identical endpoint outcomes (including inherent service errors), TOOLPRO exposes the same per-call outputs/errors as stepwise execution. Across repair-driven re-executions for the same intent instance, completed WRITE effects are not duplicated at the service boundary.

- **Safe fallback.** If construction or repair exceeds the attempt budget, violates constraints, or encounters unsupported WRITE semantics, TOOLPRO falls back to stepwise endpoint calling and surfaces diagnostics.

The contract is realized by three mechanisms, each addressing one of the three challenges (introduced in §2), as shown in Figure 2.

### 3.2. Constraint-Guided Construction with Compiler and Runtime Feedback

The first challenge to realize tool programs is to ensure reliable program execution (C1). In practice, LLM-produced programs are often structurally plausible yet fail under a typed sandbox due to unsupported libraries, type mismatches, missing imports, or brittle error handling. TOOLPRO addresses this gap with a constraint-guided construc-

tion pipeline with four steps, which turns failures into bounded, checkable repairs using compiler and runtime feedback.

**Step 1: client-side synthesis with lightweight interface checks.** Given the task intent and tool specifications, the client synthesizes a candidate program $P$ and performs conservative checks that are cheap yet effective at filtering obviously misaligned candidates. Concretely, it verifies (i) endpoint coverage (required endpoints appear), (ii) a control-flow skeleton consistent with the intent (e.g., a bounded loop and necessary conditionals), and (iii) basic value-flow sanity (outputs of earlier READ calls can be bound and used as inputs to later calls). These checks avoid sending clearly ill-formed programs into server-side compilation.

**Step 2: server-side projection into a constrained surface.** Upon receiving $P$, the server applies a deterministic projection $\Pi(\cdot)$ to rewrite it into the constrained interface-program surface. The projection (i) rewrites all external interactions to the unified stub $\text{CALL}(\cdot)$ with explicit READ/WRITE annotations, (ii) removes or replaces unsupported imports and patterns, and (iii) rejects any syntax or language features outside the allowed surface. The result, a canonical form $\Pi(P)$, ensures that subsequent diagnostics and traces are expressed over a stable surface that the runtime can interpret, enforce, and modify reliably.

**Step 3: compile and execute with feedback-driven, bounded in-place repair.** The server then compiles and executes $\Pi(P)$ in a sandbox and uses the resulting feedback to repair and re-run within a fixed attempt budget. On compilation failure, the compiler returns precise diagnostics that localize missing symbols, type mismatches, and offending code spans. On runtime failure, the runtime records a lightweight trace that identifies the failing region together with the prefix of dynamic calls already observed. TOOLPRO applies *in-place* repairs conditioned on this verifiable feedback and re-enters the loop. It re-projects the revised program if needed, recompiles, and re-executes until success or the budget is exhausted. Repairs are localized whenever possible. Invocation errors trigger edits at the relevant call site (endpoint, arguments, or effect annotation), while control-flow/logic errors trigger rewrites of the minimal affected block with the rest preserved.

**Step 4: Safe fallback with diagnostics.** If the attempt budget is exceeded or a repair violates the constrained surface, TOOLPRO falls back to stepwise endpoint calling and surfaces the diagnostics collected during steps. This ensures that program execution is attempted only when it is demonstrably viable, and that failures are surfaced with auditable diagnostics rather than silently ignored.

## 3.3. Effect-Aware Replay for Retry-Safe Semantics

The second challenge (C2) is retry safety under repair-driven re-execution. A tool program may partially succeed before failing. Naively re-running repaired code can re-emit state-modifying WRITE calls, duplicating side effects and corrupting service state. TOOLPRO prevents this by mediating runtime calls at the interface boundary and replaying outcomes of completed WRITE calls across re-executions.

Because retry safety must hold at runtime, TOOLPRO intercepts each dynamic call (a runtime instance of $\mathrm{CALL}(e, a)$) along the taken control-flow path. READ calls are always forwarded to the service, since re-issuing them does not introduce new side effects. In contrast, WRITE calls are replay-protected. Once a WRITE completes, subsequent re-executions must not re-emit it, while the program continues to observe the same outcome.

To support such replay, TOOLPRO maintains a per-intent-instance log of committed WRITE outcomes. Specifically, it keeps (i) an ordered history log $\mathcal{H}$ containing WRITE calls completed in prior executions, and (ii) a working log $\mathcal{W}$ for WRITE calls completed in the current execution. Each entry stores $(e, a, o)$ along with a per-re-execution flag used. Both logs are scoped to a single intent instance, ensuring replay does not leak across unrelated tasks.

Since replay is only sound when repairs do not attempt to revise already committed effects, TOOLPRO enforces a conservative replay discipline. If a repaired program changes the arguments or relative order of any committed WRITE prefix already recorded in $\mathcal{H}$, TOOLPRO disables replay and falls back to stepwise calling. This turns a potential silent semantic divergence into an explicit, auditable condition.

Under this discipline, replay reduces to a simple matching problem at each WRITE. On re-execution, when the program reaches the next dynamic WRITE with parameters $(e, a)$, TOOLPRO matches it to the earliest unused entry in $\mathcal{H}$ with the same $(e, a)$. If a match exists, the runtime returns the cached outcome $o$ without issuing the external call; otherwise, it emits the call, obtains $o$, and appends $(e, a, o)$ to $\mathcal{W}$. Ordered matching treats repeated writes with identical $(e, a)$ as distinct dynamic calls within a run, while still preventing duplication across re-executions.

The log-based replay mechanism is shown in Algorithm 1. Before each re-execution, the runtime archives the WRITE calls completed in the last execution by appending $\mathcal{W}$ to $\mathcal{H}$, clears $\mathcal{W}$, and resets all used flags. READ calls are always emitted, whereas WRITE calls are either replayed from $\mathcal{H}$ (if matched) or emitted and recorded into $\mathcal{W}$.

**Proposition 3.1** (Retry-safe WRITE emissions). *Fix an intent instance and a well-formed interface program $P$. Across repair-driven re-executions that satisfy the replay discipline above,* TOOLPRO *emits each completed dynamic*

---

**Algorithm 1** Effect-Aware Replay for Retry-Safe Semantics

**Require:** $\mathcal{H}$: ordered history of completed WRITE calls across prior executions
**Require:** $\mathcal{W}$: ordered log of completed WRITE calls in the current execution

1: **procedure** ONREEXECUTION
2:     $\mathcal{H} \leftarrow \mathrm{Concat}(\mathcal{H}, \mathcal{W})$     ▷ Append last execution in order
3:     $\mathcal{W} \leftarrow \emptyset$
4:     **for each** entry $x \in \mathcal{H}$ **do**
5:         $x.\text{used} \leftarrow$ **False**

6: **function** HANDLECALL$((e, a, \text{eff}))$
7:     **if** $\text{eff} = \text{READ}$ **then**
8:         **return** EMIT$((e, a))$
9:     **else**                                    ▷ $\text{eff} = \text{WRITE}$
10:         **for each** entry $x \in \mathcal{H}$ **in order do**
11:             **if** $x.(e, a) = (e, a)$ **and** $\neg x.\text{used}$ **then**
12:                 $x.\text{used} \leftarrow$ **True**
13:                 **return** $x.o$          ▷ Replay outcome
14:         $o \leftarrow$ EMIT$((e, a))$
15:         $\mathcal{W}.\text{append}((e, a, o))$
16:         **return** $o$

---

*WRITE call to the underlying service at most once, and later re-executions replay its cached outcome.*

*Proof.* The only emission of a WRITE occurs when no unused matching entry exists in $\mathcal{H}$ (Lines 9–16), in which case the emitted outcome is recorded and later archived into $\mathcal{H}$. In subsequent re-executions, the same dynamic WRITE must match an unused entry and be replayed (lines 10–13), so it cannot be emitted again. ☐

Moreover, practical services often provide idempotency keys or exhibit nondeterminism, which affects replay matching. When idempotency keys are available, TOOLPRO includes them in argument $a$, strengthening matching and aligning replay with the service's own exactly-once intent. If an endpoint is meaningfully nondeterministic under identical $(e, a)$ and no idempotency is available, TOOLPRO conservatively falls back to stepwise calling, as replay could otherwise lead to different results compared to stepwise execution.

### 3.4. Profile-Driven Consolidation Policy

The remaining challenge (C3) is *when* to pay the one-time build cost of program execution. While tool programs can reduce client-server turns, their construction (synthesis, projection, compilation, and possible repair) is nontrivial, so consolidation should be used only when it is predicted to reduce end-to-end cost. TOOLPRO therefore uses lightweight

online profiling and decision rules to choose between the tool program and stepwise calling.

To make this choice instance-adaptive, TOOLPRO maintains moving averages from recent runs for $\overline{T_{\text{RTT}}}$, $\overline{T_{\text{DEC}}}$, and $\overline{T_{\text{BUILD}}}$. Here $T_{\text{RTT}}$ is the client–service round-trip time, $T_{\text{DEC}}$ is per-step client-side decision overhead, and $T_{\text{BUILD}}$ includes program synthesis, projection, compilation, and any repair time. In addition, TOOLPRO estimates $N$, the number of dynamic endpoint invocations in the candidate program, using synthesized structure and loop bounds when available.

Given these estimates, the policy follows the cost model in Equation 1. Consolidation primarily saves $(N-1)$ additional RTTs and decision rounds, while endpoint execution time largely appears in both modes. TOOLPRO predicts the net benefit of program execution over stepwise calling as

$$\Delta T \;=\; (N-1) \cdot (\overline{T_{\text{RTT}}} + \overline{T_{\text{DEC}}}) \;-\; \overline{T_{\text{BUILD}}}. \qquad (2)$$

If $\Delta T > 0$, TOOLPRO executes the tool program; otherwise it selects stepwise calling.

Because profiles may be inaccurate at cold start, TOOLPRO bootstraps with a small number of stepwise runs to initialize $\overline{T_{\text{RTT}}}$ and $\overline{T_{\text{DEC}}}$, and enables tool program execution only when the synthesized structure is clearly multi-step (e.g., a bounded loop). Once estimates stabilize, Equation 2 is applied to make per-instance decisions.

This policy integrates naturally with the end-to-end control flow. Given an intent, TOOLPRO synthesizes a candidate program, estimates $\Delta T$, and selects the mode. If program execution is chosen, the server runs projection, compilation, and bounded in-place repair under effect-aware execution. If repair exceeds the attempt budget, violates constraints, or encounters unsupported WRITE semantics, TOOLPRO falls back to stepwise calling with diagnostics. As a result, TOOLPRO uses tool programs only when they are predicted to be beneficial in efficiency.

# 4. Experiments

## 4.1. Implementation

We instantiate TOOLPRO over MCP-style tool-facing services and implement the service-side runtime using WebAssembly (Wasm). On the client, an LLM synthesizes a tool program $P$ from the intent and tool specifications and applies lightweight interface checks; the consolidation policy selects between program mode and stepwise calling. On the server, TOOLPRO projects $P$ into the constrained interface-program surface via $\Pi(\cdot)$, compiles and executes it in a sandbox with bounded in-place repair, and enforces retry-safe WRITE semantics by mediating every dynamic CALL and replaying completed WRITE outcomes across re-executions. If projection/repair violates constraints or ex-

ceeds the attempt budget, TOOLPRO falls back to stepwise calling with diagnostics.

We use Wasm as the execution substrate because it provides (i) a strong sandbox boundary for untrusted, LLM-generated code with no ambient authority, (ii) a capability-style host interface that lets us expose only the unified CALL$(e, a, \text{eff})$ stub and mediate all side effects, and (iii) portable, low-overhead execution suitable for short-lived procedural workloads. Additional implementation details appear in Appendix A.

## 4.2. Experimental Setup

**Benchmarks.** We evaluate TOOLPRO on three widely-used open-source applications (Memos, Directus, and MinIO) from GitHub, following prior studies (Gu et al., 2025). Details on these applications are shown in Appendix B.1. Each application is exposed as an MCP-style tool-facing service with fixed endpoints, and we construct procedural workflows that require loops/conditionals and intermediate bindings.

For each application, we construct two workflows: a read-only workflow (suffix `.r`) and a read-write workflow (suffix `.w`). Each workflow is parameterized by $N$, which controls procedural length (and equals the number of endpoint invocations in stepwise execution). More details on the constructed workflows are shown in Appendix B.2. We also add supplemental realistic workflows in Appendix B.3 to stress nondeterministic retrieval, branching, coordinated writes, non-idempotent effects, and cross-service execution.

**Metrics.** We report end-to-end latency and client-side traffic volume, two widely used metrics (Gu et al., 2025). Latency measures the time to complete a workflow, including client-side LLM time, client-server communication, and server-side execution. Client-side traffic measures bytes transmitted by the client during workflow execution, including both client-server payloads and client-to-LLM prompts. For supplemental workflows, we also report task accuracy.

**Baselines and variants.** We compare TOOLPRO against the prevailing stepwise endpoint interface and include two TOOLPRO variants to isolate the effect of consolidation.

- **Stepwise MCP Web Service (MWS).** The prevailing fully stepwise baseline. At each step, the agent replans the next endpoint call $(e_i, a_i)$ given the intent, tool specs, and prior observations, leading to $N$ requests and typically $N$ LLM decision rounds with repeated tool-context transmission.
- **TOOLPRO-step.** A stepwise TOOLPRO variant (no consolidation) that replaces replanning with intent-structured guidance: it first derives a coarse call skeleton from the intent (e.g., required endpoints and loop/conditional structure), and then per step only instantiates/validates the

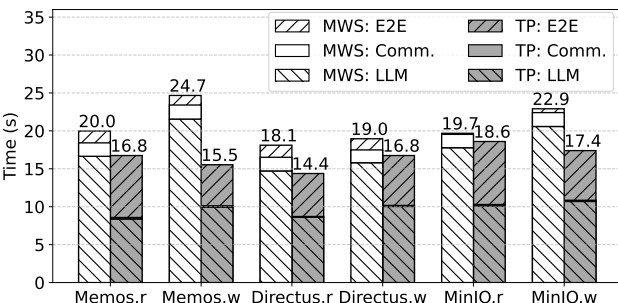

*Figure 3.* Latency comparison. 'TP' indicates TOOLPRO. 'E2E' is the end-to-end latency. 'Comm.' is client-server communication latency. 'LLM' is latency involving client-side LLM inference.

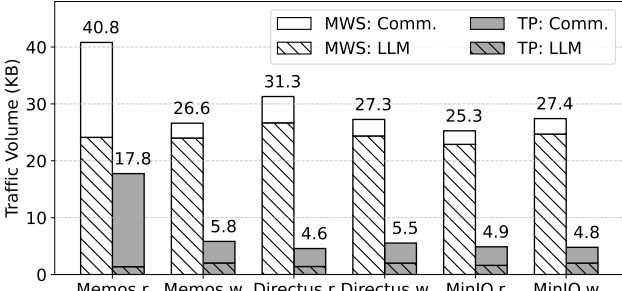

*Figure 4.* Client-side traffic volume. 'Comm.' denotes client-server bytes. 'LLM' denotes client-to-LLM bytes.

next call using current observations (e.g., filling arguments from prior READ outputs), still incurring $N$ requests/rounds but with lower per-step decision overhead and less repeated tool context.

- **TOOLPRO-prog.** TOOLPRO-prog always executes in program mode. The agent submits one effect-typed tool program, which the server projects ($\Pi(\cdot)$), compiles/repairs, and executes under call interception. This isolates the benefits of consolidation (turn reduction) while paying the one-time build cost.
- **TOOLPRO.** Full system with the profile-driven consolidation policy, selecting between program mode and stepwise mode per instance.

Details on experimental environments are shown in Appendix B.4.

### 4.3. Overall Efficiency

We first evaluate end-to-end efficiency by comparing TOOLPRO against MWS across workflows with $N = 10$. Unless specified otherwise, the server is hosted in Sydney and the client in Beijing.

**Service latency.** As shown in Figure 3, TOOLPRO consistently reduces end-to-end latency across all workflows. The main source of improvement is turn reduction: program mode packages a multi-step interaction into a single submission/execution cycle, avoiding the repeated client-side decide-next-call loop in MWS and reducing $(N - 1)$ additional RTTs and reasoning rounds. While TOOLPRO incurs a one-time build cost (program synthesis, projection, compilation, and occasional bounded repair), this overhead is amortized for procedural workflows and higher RTT settings, consistent with the proposed policy model. Latency improvements remain consistent for read-write workflows, indicating that enforcing retry-safe semantics does not dominate end-to-end cost.

**Client-side traffic volume.** As shown in Figure 4, TOOLPRO reduces client-side traffic by up to 85.3%. This reduction is primarily driven by fewer client-to-LLM exchanges: MWS repeatedly transmits tool specifications and intermedi-

ate context across $N$ rounds, whereas program mode transmits a compact tool program once (plus bounded repair prompts when needed). Client–server bytes may fluctuate because program mode uploads program source/bytecode, but the reduction in repeated prompting and over/under-fetching dominates for procedural workflows.

**Realistic workflows and reliability.** To test whether the gains go beyond fixed-$N$ procedural loops, we add four supplemental complex benchmarks (cbench1–cbench4) that require nondeterministic retrieval, loop/branch logic, coordinated multi-record writes, non-idempotent side effects, and cross-service branching. Across cbench1–cbench3 with complex realistic workflows, TOOLPRO reduces end-to-end latency from 30.16s to 17.91s (40.6%), improves task accuracy from 0.60 to 0.93, and reduces client-side LLM latency from 14.98s to 7.08s (52.8%). On the cross-service benchmark (cbench4), TOOLPRO reduces latency from 52.68s to 24.54s (53.4%), cuts client-side traffic by 96.1%, and improves accuracy from 0.20 to 0.80.

We observe that the program-mode failure rate is relevant to the coding capability of LLMs. On the Rust-based cbench2, qwen3-coder-flash reaches 80% success rate, while gpt-5.1 and gemini-3-flash-preview each reach 100% success rate with no observed compilation failure or fallback. In addition, replay also matters operationally: over a 15-run no-replay ablation on cbench1–cbench3, disabling replay increases average latency from 17.92s to 21.45s (+19.7%) and fallback from 0/15 to 3/15. Thus TOOLPRO fails closed when replay safety cannot be guaranteed, and fallback does not erase the overall gains over complex realistic workflows.

### 4.4. Sensitivity

We next study how network conditions and workflow complexity affect performance, using `Memos.w`.

**Impact of network conditions.** To vary network conditions, we deploy servers in London, Sydney, and San Francisco with a client in Beijing, and also inject additional one-way delays of 50ms, 100ms, 150ms, and 200ms. We run with $N = 10$.

As shown in Figure 5, TOOLPRO outperforms MWS across

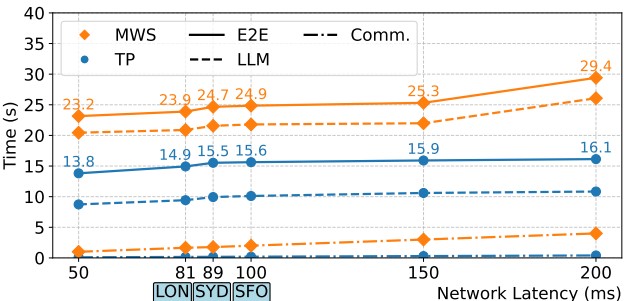

*Figure 5.* Impact of varying network conditions.

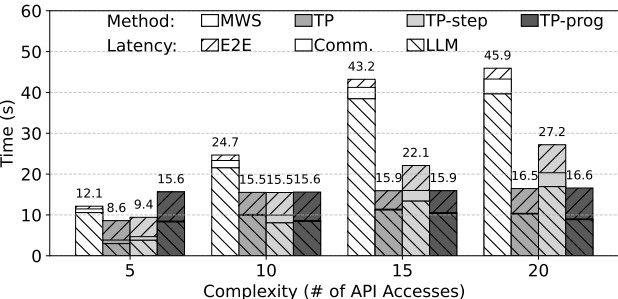

*Figure 6.* Impact of workflow complexity on latency.

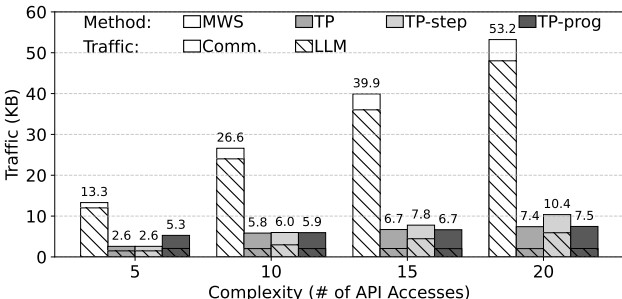

*Figure 7.* Impact of workflow complexity on traffic volume.

$N$ due to repeated prompts and tool-context transmission. TOOLPRO-prog maintains low traffic for larger $N$ by avoiding multi-round prompting. By switching between stepwise and program modes, TOOLPRO minimizes client-side traffic across complexity conditions.

Moreover, we break down the overhead of TOOLPRO-prog in Appendix C.

## 5. Related Work

**Interfaces and representations for agent tool use.** LLM-based agents increasingly solve tasks by invoking tools and APIs (Shen et al., 2025a; Liu et al., 2026; Shen et al., 2025b; Du et al., 2024; Song et al., 2025; Schick et al., 2023; Qin et al., 2024). The dominant interface remains stepwise endpoints, where intent is implicit in a sequence of endpoint calls interleaved with multi-round reasoning. This makes control flow and intermediate bindings an emergent property of the agent policy, inflating network turns and reasoning as workflows lengthen. In contrast, TOOLPRO makes intent explicit as an executable tool program with a constrained surface and effect types, enabling compilation, inspection, enforcement, and optimization at the interface boundary.

**From expressive queries to executable interaction logic.** Web APIs evolved from RESTful endpoints (Fielding, 2000) to flexible query interfaces such as GraphQL (GraphQL, 2015), reducing over/under-fetching by letting clients shape responses. However, GraphQL is not designed to encode procedural interaction logic (loops, conditionals, retries) as a single interface object (Stack-Overflow, 2018a; Hartig & Pérez, 2018; Stack-Overflow, 2018b). Recent systems execute user-supplied logic near the service; e.g., ORFA (Gu et al., 2025) uses Wasm modules as a Turing-complete query language. TOOLPRO instead focuses on agentic workflows where the core bottlenecks are executability and side effects under re-execution, using LLM-synthesized, repairable tool programs with an effect-typed contract and retry-safe semantics under iterative repair.

**Programmatic agents and in-situ execution.** Code-based agent methods such as CodeAct (Wang et al., 2024) use

network settings. As RTT increases, the performance gap widens because consolidation saves $(N-1)$ additional round trips, and the policy increasingly favors program mode when the predicted benefit exceeds build cost. In low-latency conditions, the policy more often selects stepwise mode to avoid paying the one-time build cost, yielding robust performance across conditions. An extended 1–2000ms RTT sweep on `Memos.w` ($N=8$) makes the mode switch explicit. TOOLPRO-step is better at 1–100ms (14.50–15.30s versus 15.51–15.60s for TOOLPRO-prog), while TOOLPRO-prog is better at 1000–2000ms (16.50–17.51s versus 22.48–30.50s for TOOLPRO-step). The full TOOLPRO policy stays close to the empirically better mode across the sweep, selecting stepwise execution in low-latency settings and program execution as RTT dominates.

**Impact of workflow complexity.** We vary workflow complexity by setting $N = 5, 10, 15, 20$ and compare MWS, TOOLPRO-step, TOOLPRO-prog, and TOOLPRO, with the server hosted in Sydney and the client in Beijing.

As shown in Figure 6, MWS scales roughly linearly with $N$ due to $N$ rounds of RTT and reasoning. TOOLPRO in stepwise mode improves modestly by reducing per-step decision overhead with intent-structured guidance, but still pays $N$ rounds. In contrast, TOOLPRO-prog remains comparatively stable with $N$ because it pays a one-time build cost and executes the interaction logic server-side. The full TOOLPRO achieves the best of both by switching between modes, validating the profile-driven policy.

As shown in Figure 7, client-side traffic in MWS grows with

executable code as an agent action space, and workflow-generation methods such as AFlow (Zhang et al., 2025a) search over code-represented agent workflows. TOOLPRO is complementary but targets a different boundary: the submitted program is an effect-typed service interface object, not only an internal reasoning/action representation, and the runtime must enforce replay-safe WRITE behavior across repair and re-execution. The design is also analogous to in-situ programmable execution systems such as eBPF (Hoiland-Jorgensen et al., 2018), which move logic closer to the execution boundary to reduce repeated control transfers. Unlike eBPF's pre-verified kernel packet programs, TOOLPRO handles dynamically generated tool programs over web-service endpoints with explicit effect annotations, sandboxed execution, and fail-closed fallback.

**Wasm sandboxing as a substrate.** WebAssembly (Wasm) provides a portable sandbox with near-native performance (Liu et al., 2025b; Yan et al., 2021; Liu et al., 2025a) and is widely adopted beyond browsers, including serverless and edge settings (Hoque & Harras, 2023; Ménétrey et al., 2022; Kjorveziroski & Filiposka, 2023; Gackstatter et al., 2022; Zhang et al., 2025b). TOOLPRO uses Wasm as a supporting substrate to securely execute the constrained tool-program surface.

## 6. Conclusion

We introduced TOOLPRO, which makes *tool programs* a first-class agent-facing service interface. By compiling effect-typed programs with projection and bounded repair, TOOLPRO consolidates multi-step workflows into fewer turns while ensuring retry-safe execution via effect-aware replay. A profile-driven policy selects program execution only when it is predicted to reduce end-to-end cost. Across complex workflows within realistic applications, TOOLPRO reduces latency and client-side traffic, with gains growing under higher RTT and increased workflow complexity. We believe TOOLPRO is a practical step toward executable and effect-safe representations of tool intent for the agentic web.

## Acknowledgements

This work was supported by National Natural Science Foundation of China under the grant number 62595734, the Key Laboratory of High Confidence Software Technologies (Peking University), the Ministry of Education, and the Center for Data Space Technology and System, Peking University.

## Impact Statement

This paper presents work whose goal is to advance the field of Machine Learning. There are many potential societal consequences of our work, none of which we feel must be specifically highlighted here.

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

# A. Implementation Details

This appendix provides additional details of TOOLPRO's Wasm-based runtime, including the host interface, replay state management, and policy instrumentation.

## A.1. Runtime Architecture

TOOLPRO is a client-server system. The client produces a candidate tool program $P$ in Rust and an intent-instance identifier, and then either (i) submits $P$ to the server for program-mode execution or (ii) performs stepwise calling. In program mode, the server executes the following stages. (1) *Projection* $\Pi(\cdot)$ rewrites $P$ into the constrained interface-program surface. (2) *Compilation/repair* utilizes Rustc to compile the projected Rust program and performs bounded in-place repair using compiler diagnostics and runtime traces. (3) *Sandboxed execution* runs the resulting module while intercepting every dynamic CALL to enforce program order and effect-aware replay. If any stage exceeds budgets or violates constraints, the server triggers safe fallback to stepwise calling and returns diagnostics.

## A.2. Wasm Runtime Configuration and Host Interface

The overall Wasm generation pipeline is shown in Figure 8. TOOLPRO compiles the LLM-generated Rust program into a Wasm module and executes it using Wasmtime. The module runs with no ambient authority. It cannot issue network requests, access the filesystem, or load dynamic libraries. All external interaction is mediated through a minimal set of host imports.

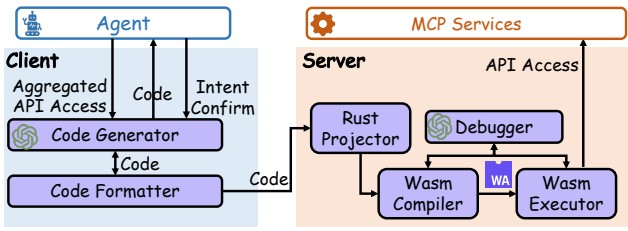

*Figure 8.* Wasm generation pipeline of TOOLPRO.

**Unified call stub.** The only capability needed by tool programs is the unified stub $\text{CALL}(e, a, \text{eff})$. We implement this as a host-imported function that takes (i) an endpoint identifier $e$ (interned string or integer id), (ii) a serialized argument object $a$ (e.g., JSON bytes), and (iii) an effect tag $\text{eff} \in \{\text{READ}, \text{WRITE}\}$. The host returns a serialized result object or a typed error. This import is the enforcement point. The mediator can log calls, enforce program order, check effect labels, and apply replay semantics without requiring the module to embed networking logic.

**Deterministic projection target.** Projection $\Pi(\cdot)$ targets a stable surface that compiles deterministically to Wasm. In

particular, it rewrites all external interactions to the stub, rejects disallowed constructs (exceptions/threads/dynamic linking/unsafe features), and ensures effect annotations are explicit at each call site. This makes compiler diagnostics and runtime traces comparable across repair iterations.

## A.3. Effect Mediation and Replay State

Effect-aware replay is implemented inside the mediator that handles every dynamic CALL from the Wasm module. For each intent instance, the mediator maintains (i) a history log $\mathcal{H}$ of completed WRITE calls from prior executions, and (ii) a working log $\mathcal{W}$ for completed WRITE calls in the current execution. Each entry stores $(e, a, o)$ and a per-run `used` flag.

**Re-execution protocol.** Before each repair-driven re-execution, the mediator archives $\mathcal{W}$ into $\mathcal{H}$, clears $\mathcal{W}$, and resets all `used` flags. During execution, READ calls are always forwarded to the service. For a WRITE call with parameters $(e, a)$, the mediator matches it to the earliest unused entry in $\mathcal{H}$; if matched, it returns the cached outcome without emitting the call, otherwise it emits once and appends the outcome to $\mathcal{W}$.

**Fail-closed conditions.** Replay is enabled only under the replay discipline. If a repaired program attempts to revise a committed WRITE prefix (by changing arguments or relative order), replay is disabled, and the system falls back to stepwise calling with diagnostics.

# B. Experimental Details

## B.1. Application Benchmarks

Details of employed applications are listed below.

- **Memos**[2] (56k stars): A lightweight, self-hosted knowledge management platform in Go, exposing OpenAPI-compliant REST interfaces.
- **Directus**[3] (34k stars): An API layer providing REST and GraphQL endpoints over SQL backends.
- **MinIO**[4] (60k stars): An S3-compatible, high-performance object storage system.

## B.2. Constructed Workflows

Table 1 shows the constructed procedural agentic workflows used in our experiments.

---

[2] https://github.com/usememos/memos
[3] https://github.com/directus/directus
[4] https://github.com/minio/minio

*Table 1.* Composed procedural workflows over fixed endpoints.

| App | Read-Only Workflow (.r) | Read-Write Workflow (.w) |
|---|---|---|
| Memos | Get memos of $N$ users (user_id=1,2,…,$N$). $N$ is the user number. | Change the visibility of $N$ memos. $N$ is the memo number. |
| Directus | Get detailed information of $N$ articles (id=1,2,…,$N$). $N$ is the article number. | Modify author information of $N$ draft articles. $N$ is the number of articles. |
| MinIO | Download $N$ files. $N$ is the file number. | Upload $N$ files. $N$ is the file number. |

## B.3. Supplemental Realistic Workflows and Results

The supplemental realistic benchmarks were added to measure TOOLPRO under complex situations. They keep the same MCP-style endpoint setting but require runtime binding, branch-dependent control flow, state consistency across multiple records, and cross-service side effects.

*Table 2.* Supplemental realistic workflows.

| Bench | Core challenge | Representative task |
|---|---|---|
| cbench1 | Nondeterministic retrieval | Find customer Mei Patel by ZIP 76165, identify the correct pending order at runtime, and update that order to contain only item 1096508426 with total 55.0. |
| cbench2 | Loop, branching, and coordinated writes | Update all pending orders and the customer's primary address; if order #W4082615 has not shipped, replace its items and set the total to 25.0. |
| cbench3 | Non-idempotent side effects | For delivered order #D12345, change status to exchange_requested, replace its items, and create an exchange log recording old and new items. |
| cbench4 | Cross-service branching | Upload five local documents to MinIO, inspect object text, route objects across prefixes, create follow-up memos when required, then clean up generated files and memos. |

**Supplemental benchmark results.** Across cbench1–cbench3, TOOLPRO reduces average latency from 30.16s to 17.91s (40.6%), improves task accuracy from 0.60 to

0.93, and lowers client-side LLM latency from 14.98s to 7.08s. For cbench4, TOOLPRO reduces latency from 52.68s to 24.54s (53.4%), cuts client-side traffic by 96.1%, and improves accuracy from 0.20 to 0.80. The policy selects program mode in 12/15 cold-start runs and in all 12/12 warm-start runs after profiles are available; no fallback is observed once program mode is selected in these measurements.

**Replay and model comparison.** In a 15-run no-replay ablation on cbench1–cbench3, disabling replay increases average latency from 17.92s to 21.45s and fallback from 0/15 to 3/15. On cbench2, gpt-5.1 and gemini-3-flash-preview each reach 100% success rate with no observed compilation failure or fallback, while qwen3-coder-flash reaches 80%, indicating that compilation failures are primarily a model-capability bottleneck.

*Table 3.* Extended RTT sweep on `Memos.w` with $N = 8$.

| RTT | TOOLPRO-step | TOOLPRO-prog | TOOLPRO policy |
|---|---|---|---|
| 1ms | 14.50s | 15.51s | 14.54s |
| 10ms | 14.57s | 15.52s | 14.57s |
| 100ms | 15.30s | 15.60s | 15.43s |
| 1000ms | 22.48s | 16.50s | 16.22s |
| 2000ms | 30.50s | 17.51s | 17.64s |

This sweep makes the profile-driven switch explicit: stepwise execution is preferable at low RTT, while program execution dominates once RTT becomes the main cost.

*Table 4.* Language comparison on cbench2. Rust is an implementation choice rather than a conceptual requirement.

| Language | Avg. seconds | Accuracy | Note |
|---|---|---|---|
| Rust | 19.4052 | 0.80 | Mature direct-to-Wasm frontend in our setting. |
| Go | 50.2153 | 0.40 | More brittle direct-to-Wasm compilation in our setting. |
| Lua | N/A | N/A | No practical direct-to-Wasm path for this use case. |

These results support using Rust in the prototype because it integrates cleanly with Wasmtime and provides actionable compiler diagnostics, while a lighter DSL/IR remains a promising future direction.

Generated tool programs remain compact in the supplemental runs: sampled programs have a median of 70 non-empty lines of code, 4 MCP tool calls, and 1 branch/loop keyword hit; the shortest and longest samples are 43 and 102 non-empty lines. The 102-line sample includes 6 tool calls (3 READ, 3 WRITE) plus a conditional branch, with explicit effect annotations at call sites.

## B.4. Environments

We run servers on AWS `c7i-flex.large` instances (Intel Xeon Sapphire Rapids @2.40GHz, 2 vCPUs, 4GB RAM, up to 12.5Gbps). Servers are hosted in Sydney, London, and San Francisco. The client runs in Beijing on a machine with an Intel Core i9-14900HX CPU, 16GB RAM, and a gigabit network connection. TOOLPRO is implemented in

Python, utilizing the Wasmtime[5] for WebAssembly execution. TOOLPRO uses Rust[6] as the high-level language for Wasm compilation. We use Qwen3-Max-Instruct (Qwen-Team, 2025) for program synthesis and repair prompting on the client. Unless otherwise specified, results are averaged over five runs.

## C. Overhead Breakdown

*Table 5.* Breakdown of tool-program build pipeline latency (seconds and percentage of total end-to-end latency).

| App | $N$ | Program Synthesis | Compilation | Sandbox Execution | Service Execution |
|---|---|---|---|---|---|
| Memos.r | 5 | 7.31 (41.9%) | 5.39 (30.9%) | 2.44 (14.0%) | 2.06 (11.8%) |
| | 10 | 7.32 (40.9%) | 5.43 (30.3%) | 2.46 (13.7%) | 2.40 (13.4%) |
| | 15 | 7.61 (41.9%) | 5.39 (29.7%) | 2.45 (13.5%) | 2.58 (14.2%) |
| | 20 | 7.61 (41.3%) | 5.42 (29.4%) | 2.47 (13.4%) | 2.81 (15.2%) |
| Memos.w | 5 | 7.31 (42.4%) | 5.17 (30.0%) | 2.47 (14.3%) | 2.16 (12.5%) |
| | 10 | 7.51 (42.0%) | 5.21 (29.2%) | 2.48 (13.9%) | 2.20 (12.3%) |
| | 15 | 7.62 (42.3%) | 5.37 (29.8%) | 2.47 (13.7%) | 2.38 (13.2%) |
| | 20 | 7.86 (41.3%) | 5.45 (28.6%) | 2.49 (13.1%) | 3.09 (16.2%) |

We break down the overhead of program mode by measuring TOOLPRO-prog on `Memos.r` and `Memos.w` with $N = 5, 10, 15, 20$ (server in Sydney, client in Beijing). Table 5 shows that program synthesis dominates the one-time build cost, while compilation and sandbox execution remain stable across $N$. This suggests that model specialization for the constrained interface-program surface (or more efficient synthesis/repair prompting) could further reduce overhead. Importantly, sandbox execution includes effect mediation for WRITE calls, yet contributes a relatively small fraction of end-to-end time, indicating that retry safety can be enforced with modest overhead in practice.

---

[5] `https://github.com/bytecodealliance/wasmtime`
[6] `https://rust-lang.org/`

