# OpenReview forum: "Beyond Static Endpoints: Tool Programs as an Interface for Flexible Agentic Web Services"
_ICML.cc/2026/Conference — ICML 2026 regular_

### Official Review · Reviewer_hgUu · 2026-03-12

**Soundness:** 3
**Presentation:** 3
**Significance:** 3
**Originality:** 3
**Overall Recommendation:** 5
**Confidence:** 3

**Summary:**

This paper addresses a critical bottleneck in how LLM-based agents interact with web services, namely that static endpoints force stepwise, brittle multi-round interactions that inflate latency and network traffic. The authors introduce tool programs, an executable representation of multi-step service intent with explicit READ/WRITE effect annotations, and present TOOLPRO, a system that synthesizes, compiles, and executes these programs in a WebAssembly sandbox. TOOLPRO ensures retry-safe semantics under partial failures, consolidates multi-step interactions to reduce client-side decision rounds, and uses a profile-driven policy to adaptively choose between program execution and stepwise endpoint calls. Experiments on three open-source applications and six workflows demonstrate up to 37.2% latency reduction and 85.3% reduction in client-side traffic, with gains increasing under high network latency and complex workflows.

**Compliance With Llm Reviewing Policy:**

Affirmed.

**Final Justification:**

The authors have fully addressed my concerns.

**Key Questions For Authors:**

- How would TOOLPRO handle workflows with dynamically discovered endpoints or unpredictable branching that cannot be fully anticipated during program synthesis?
- Have you considered integrating fallback mechanisms for nondeterministic endpoints beyond stepwise execution, especially when idempotency keys are unavailable?
- Could the authors clarify how TOOLPRO compares to related approaches such as CodeAct, CRAFT, or Agent Skill Induction, specifically in terms of (i) the role of executable code, (ii) internal vs external abstraction, (iii) learning vs execution focus, and (iv) handling side effects and workflow consolidation?

**Limitations:**

yes

**Strengths And Weaknesses:**

# Strengths

- The paper identifies a real representational bottleneck in multi-step agentic workflows by showing that static endpoint sequences fragment coherent plans into brittle stepwise calls, which inflates latency and network usage; the experimental results on Memos, Directus, and MinIO workflows provide concrete evidence of up to 37% latency reduction and 85% traffic savings when using TOOLPRO’s consolidated tool programs.
- TOOLPRO introduces effect-aware replay that enforces exactly-once semantics for WRITE operations, ensuring safe re-execution under failures, and the implementation details, including per-intent logging of completed WRITE calls, demonstrate a robust mechanism for managing side effects without duplicating state.
- The system’s design, including a constrained program surface with structured control flow and server-side execution in a Wasm sandbox, allows safe execution of LLM-generated programs while reducing client-side decision overhead, and the latency breakdown figures show that this contributes to meaningful end-to-end efficiency gains.
- The profile-driven consolidation policy is evidence-based and adaptive, balancing program synthesis cost against potential round-trip savings, which is validated by experiments under varying network latencies and workflow complexities, showing the system intelligently switches between program and stepwise execution.

# Weaknesses

- Missing discussion on related work around agent generating code as action/tools, e.g., https://arxiv.org/abs/2504.06821, https://arxiv.org/abs/2402.01030, https://arxiv.org/abs/2309.17428 -- To my understanding, this work focus on a different layer of the problem (i.e., the server side of the tool call), where previous work might focus on different layers, but it'll still be useful to include some discussion in the paper to help clarify this.

- TOOLPRO depends on LLM-produced Rust code compiling to Wasm, and the reported 20% program-mode failure rate highlights that compilation and repair can still be a significant source of runtime overhead, which may limit generalizability to other languages or more complex services.

---

> ### Author Rebuttal · Authors · 2026-03-31
>
> Dear Reviewer hgUu,
>
> Thank you for your constructive feedback and careful reading of our work. We address your questions and suggestions below and hope our responses help. We are happy to provide further clarification if needed.
>
> - **W1 & Q3. Related-work discussion.**
>
> CodeAct/CRAFT/Agent Skill Induction mainly use code as an internal planning or skill medium, whereas TOOLPRO uses an effect-typed executable tool program as the action representation itself. The comparison is:
>
> | Axis | CodeAct / CRAFT / Agent Skill Induction | TOOLPRO |
> | --- | --- | --- |
> | Role of executable code | Code is mainly a planning or skill medium | Code is the emitted action representation itself |
> | Internal vs external abstraction | Richer internal agent-side control abstraction | External, effect-typed executable tool program over the API surface |
> | Learning vs execution focus | Emphasis on capability acquisition or policy improvement | Emphasis on executability, replay-safe repair, and cost-aware mode selection |
> | Side effects and consolidation | Not primarily centered on side-effectful external actions or workflow consolidation | Explicit READ/WRITE effects, replay-safe re-execution, and consolidation of many local next-call decisions into one executable program |
>
> This also clarifies why TOOLPRO differs both from prior code-based agent work and from the eBPF analogy raised elsewhere. Relative to CodeAct/CRAFT/Agent Skill Induction, the key change is the execution-time action interface, not just the reasoning substrate. Relative to eBPF, the commonality is moving logic closer to execution, but the setting is fundamentally different: eBPF targets deterministic kernel execution, whereas TOOLPRO targets agent-generated, dynamic, side-effectful web-service interaction programs. That is why explicit effect typing, replay-safe repair, and fail-closed fallback are central here.
>
> Empirically, this difference is not merely a communication effect: across bench1-3, TOOLPRO raises mean accuracy from 0.60 to 0.93 while reducing mean client-side llm latency from 14.98s to 7.08s (-52.8%). On bench2 (baseline vs TOOLPRO), client-side llm latency drop from 24.15s to 7.50s, accuracy rises from 0.6 to 0.8, and client-side traffic drops from 100,902 bytes to 5,149 bytes (-94.9%). On cross-service bench4, TOOLPRO also reduces latency from 52.68s to 24.54s and improves accuracy from 0.2 to 0.8. This is consistent with the representational claim: one executable tool program can reduce repeated next-call decisions and client-side context/memory accumulation of agent.
>
> - **W2. Dependence on compilable Rust-to-Wasm programs and the reported failure rate.**
>
> Rust-to-Wasm is our implementation choice. We have added experiments to solve these concerns. Please see our responses to Reviewer `3N7o`, W3 and Reviewer `BKQz`, Q1.
>
> - **Q1. Dynamically discovered endpoints or unpredictable branching.**
>
> Within a known endpoint set, TOOLPRO already supports data-dependent branching and runtime discovery of entities/arguments inside the generated program; this is exactly what the more realistic benchmarks summarized in the combined reply to Reviewer `3N7o`, W2/Q2/Q3 are designed to test.
>
> - **Q2. Nondeterministic endpoints when idempotency keys are unavailable.**
>
> Our current answer is deliberately conservative: when replay soundness cannot be guaranteed, TOOLPRO disables replay and falls back to stepwise execution rather than overstating guarantees. The replay-overhead evidence is summarized in Reviewer `BKQz`, W2, and the failure-taxonomy evidence in Reviewer `3N7o`, W3.

---

> > ### Author Rebuttal · Reviewer_hgUu · 2026-04-03
> >
> > Thank you for the detailed and well-structured response. I will raise my score and vote for acceptance

---

> > > ### Author Response · Authors · 2026-04-07
> > >
> > > We sincerely thank you for your supportive review. We are glad that our rebuttal addressed your concerns, and we really appreciate your positive assessment.

---

### Official Review · Reviewer_BKQz · 2026-03-13

**Soundness:** 2
**Presentation:** 3
**Significance:** 3
**Originality:** 3
**Overall Recommendation:** 3
**Confidence:** 4

**Summary:**

This paper proposes ToolPro, a system that allows LLM agents to submit tool programs instead of issuing multiple stepwise endpoint calls. On the server side, it executes these programs in a sandboxed runtime with mechanisms for safe WRITE handling and a policy that decides when to switch to program execution from stepwise calls. Empirical results over several real applications shows end-to-end latency and client-side traffic reduction, as compared to the traditional stepwise tool invocation.

**Compliance With Llm Reviewing Policy:**

Affirmed.

**Final Justification:**

The authors have sufficiently addressed my concerns with their follow up experiments.

**Key Questions For Authors:**

1. Why generate Rust? It takes longer to compile than some other languages, and it’s not clear that the restrictiveness of its ownership system is helpful for orchestrating tool calls. Why not generate and execute, say, Golang, Lua, or even a DSL designed for tool scripting?

2. Can you show some examples of the Rust programs that ToolPro generates? How large/complicated are they?

3.  Do you have any benchmarks that are more complex, e.g., where the tool program isn’t expressible as a simple for-loop?

**Limitations:**

See review.

**Strengths And Weaknesses:**

Strengths:
I like the general idea, and the work demonstrates measurable improvements in both inference cost and roundtrip latency.


Weaknesses:
1. The core idea is conceptually close to existing mechanisms such as GraphQL or server-side batching. The novel aspect here appears to be that LLM agents can automatically synthesize the executable workflows without adhering to fixed workflow schemas. However, the evaluation does not clearly demonstrate this advantage: the benchmarks largely consist of fetching or modifying a fixed number N of resources, which could easily be implemented as fixed batch endpoints. As a result, the experiments mainly demonstrate the well-known latency benefits of reducing network round trips, rather than validating a uniquely agentic capability.

2. The paper states that most failures are due to compilation errors, so it’s not clear that ToolPro benefits much from its replay mechanism, which must carry some non-zero cost. It would be helpful to evaluate ToolPro without any replay, and thus without the overhead of intercepting WRITE calls.

3. It’s hard to tell from Figure 5 how TP prefers the step-wise mode in low-latency conditions. In fact, the figure hardly says anything other than “TP is faster than MWS” across all the entire range of network latencies. Would presenting that figure with TP-step and TP-prog, with a wider range of network latencies (e.g., 1ms—2000ms) better illustrate the mode switch?

4. This core insight of this paper is reminiscent of eBPF in the Linux kernel to reduce the number of system calls required to accomplish some workload. It would be nice to see a comparison to that in the related work.

---

> ### Author Rebuttal · Authors · 2026-03-31
>
> Dear Reviewer BKQz,
>
> Thank you for your insightful feedback. We address your concerns below and hope our responses help clarify the contributions of our work. We are happy to provide further clarification if needed.
>
> - **W1 & Q3. Relation to GraphQL/server-side batching; benchmarks beyond simple for-loop patterns.**
>
> This concern is exactly why we added the four supplemental benchmarks and their results summarized in Reviewer `3N7o`, W2/Q2/Q3. They go beyond fixed-N batch structure by requiring nondeterministic retrieval, branching, coordinated side-effecting writes, and in bench4 a cross-service content-processing workflow. The gains come from reducing decision complexity rather than merely communication overhead. This is also the key distinction from GraphQL/server-side batching: TOOLPRO does not rely on a fixed schema of predeclared workflow families, but emits an executable, effect-typed tool program over the existing API surface.
>
> - **W2. Replay overhead and whether it materially helps.**
>
> Replay is primarily a **safety** mechanism for repair-driven re-execution. We have added an ablation study, which shows that it matters operationally. Specifically, over a 15-run no-replay ablation on bench1-3 (n=5 each), average latency worsens from 17.92s to 21.45s (+19.7%) and fallback increases from 0/15 (with-replay) to 3/15 (no-replay). Replay itself is lightweight when triggered: across 6 replay events, the measured replay handling time averages 0.89s. The larger penalty comes from replay-disabled fallback, especially on bench2 (overall latency 19.41s -> 26.48s after replay disabled, +36.4%). Unsafe rerun incidence is 0/15 in these runs.
>
> | Replay ablation | With replay | No replay |
> | --- | --- | --- |
> | Average end-to-end latency over bench1-3 | 17.92s | 21.45s (slower due to fallback) |
> | Fallback | 0/15 | 3/15 |
> | Unsafe rerun incidence | 0/15 | 0/15 |
> | Replay handling time | 0.89s avg over 6 events | N/A |
>
> - **W3. Figure 5 does not clearly show mode switching.**
>
> We have added an extended latency sweep (memo.w, N=8) to make the switch clearer. In the new 1-2000ms RTT sweep, TP-step is preferred at low latencies (1-100ms), while TP-prog is preferred at higher latencies (500-2000ms), with the crossover lying between 100ms and 500ms. Across this sweep, the full TOOLPRO policy remains close to the empirically better mode.
>
> | RTT | TP-step | TP-prog | TOOLPRO policy |
> | --- | --- | --- | --- |
> | 1ms | 14.50s | 15.51s | 14.54s |
> | 10ms | 14.57s | 15.52s | 14.57s |
> | 100ms | 15.30s | 15.60s | 15.43s |
> | 1000ms | 22.48s | 16.50s | 16.22s |
> | 2000ms | 30.50s | 17.51s | 17.64s |
> - **W4. Relation to eBPF.**
>
> This is an insight analogy. Both approaches move logic closer to the execution boundary to reduce repeated control transfers. However, eBPF operates on pre-defined, verified programs, whereas TOOLPRO targets agent-generated, dynamically constructed programs with explicit side-effect handling. For a more detailed contrast, please see our response to Reviewer `hgUu`, W1/Q3.
>
> - **Q1. Why Rust?**
>
> We added a language comparison. Note that Rust is an implementation choice rather than a conceptual requirement. We use Wasm because its **portability and sandboxing properties** are well-suited to server-side execution of untrusted generated code.
>
> | Language | Workflow | Average seconds | Accuracy | llm_seconds | Note |
> | --- | --- | --- | --- | --- | --- |
> | Rust | bench2 | 19.4052 | 0.8 | 7.4953 | Strongest practical Wasm frontend in our setting |
> | Go | bench2 | 50.2153 | 0.4 | 39.5479 | More brittle direct-to-Wasm compilation in our setting |
> | Lua | bench2 | N/A | N/A | N/A | No practical direct-to-Wasm path for this use case |
>
> In the current implementation, Rust was the most practical frontend because it has a mature Wasm toolchain, integrates cleanly with Wasmtime, and provides useful compiler diagnostics for the constrained projected surface we enforce before execution and repair. We do not view Rust as necessary in principle; a lighter DSL/IR is a reasonable future direction.
>
> - **Q2. Examples and complexity of generated programs.**
>
> We measured generated-program complexity. In our sampled programs, the median size is 70 non-empty LOC, with 4 MCP tool calls and 1 branch/loop keyword hit by our simple structural count; the shortest and longest examples are 43 and 102 LOC. The 102-LOC example contains 6 tool calls (3 READ, 3 WRITE) plus a conditional branch, and the generated programs carry explicit effect annotations at each call site. See https://anonymous.4open.science/r/code_example_for_26icml-7428 for code examples.

---

> > ### Author Rebuttal · Reviewer_BKQz · 2026-04-04
> >
> > Thank you for your rebuttal. I appreciate the new results and clarifications, and I will raise my score. I just have a couple of additional comments.
> >
> >
> > For the programming language comparison, you do not need a direct Lua-to-Wasm compiler. Instead, you can run the Lua interpreter in the Wasm environment (and you could do the same for other scripting languages). And of course it is also possible to build your own sandbox environment around a language interpreter, though that may be out of the scope of this paper—as you said, the choice of language and runtime environment is just an implementation detail.
> >
> >
> > For the comparison with eBPF, I don’t think the contrast with TOOLPRO is as great as you describe, though this may be a difference of viewpoint. My view is that, from the perspective of the OS kernel, it does not matter whether the eBPF was compiled from a static program or dynamically generated by some process. In fact, you can even use an LLM to generate C code, compile it to eBPF, and feed that to the OS kernel. So, I’m not convinced that there is a fundamental difference here, though that does not take away from the fact that your paper presents a creative application of a good idea.

---

> > > ### Author Response · Authors · 2026-04-07
> > >
> > > Thank you for your encouraging feedback and positive assessment of our work. We appreciate your insightful comments on interpreter-based sandboxing, which is a valuable direction for future work. The eBPF analogy also offers a useful perspective, and we appreciate this comparison.
> > >
> > > We noticed that the **confidence score has been increased while the overall score remains unchanged**, and just wanted to check whether this reflects your intended update. We would be happy to further clarify or address any remaining concerns if helpful.
> > >
> > > Thank you again for your effort in reviewing this paper.

---

### Official Review · Reviewer_3N7o · 2026-03-13

**Soundness:** 2
**Presentation:** 2
**Significance:** 2
**Originality:** 2
**Overall Recommendation:** 3
**Confidence:** 4

**Summary:**

This paper proposes TOOLPRO, a runtime that lets an agent submit a multi step tool program, instead of issuing stepwise endpoint calls. The paper argues that static endpoints are a representational bottleneck for long horizon workflows, and it introduces three main components, constraint guided program construction, effect aware replay for WRITE calls, and a profile driven consolidation rule that switches between program mode and stepwise mode. The experiments on three applications and six workflows show lower end to end latency, up to 37.2 percent, and lower client side traffic, up to 85.3 percent, especially when RTT and workflow length increase.

**Compliance With Llm Reviewing Policy:**

Affirmed.

**Key Questions For Authors:**

Questions

1. What is the precise novelty claim relative to prior code based or workflow based agent methods that also externalize control flow?

2. How often does TOOLPRO improve performance once fallback cost is included over a large set of mixed easy and hard tasks, not only constructed procedural workflows?

3. Can the authors evaluate more realistic service operations where WRITE semantics are more complex, or where endpoint behavior is nondeterministic and idempotency keys are unavailable?

**Limitations:**

yes

**Strengths And Weaknesses:**

Strengths

1. The paper studies a real systems problem. The claim that stepwise endpoints force repeated client side reasoning and repeated round trips is sensible, and the motivating example is easy to follow. An important concept studied by this article is that tool intent can be made explicit as a program level interface object, rather than being left implicit in a long sequence of local calls.

2. The method has a clean structure. The three parts fit together well, namely constrained synthesis and repair, replay protection for WRITE calls, and an adaptive consolidation policy. The effect aware replay design is the most interesting part, because it tries to make re execution safe at the interface boundary.

3. The empirical trend is consistent with the stated cost model. The gains get larger when latency is higher and when the number of API accesses grows, which supports the main efficiency argument.

Weaknesses

1. The central idea feels too close to existing programmatic plan and act work. The paper frames tool programs as a new interface, but much of the practical gain seems to come from consolidating many calls into one executable object. That makes the work look closer to AFlow, CodeAct, and related program based agent designs than the paper admits. The related work section itself already places this paper near GraphQL style expressivity and ORFA style executable logic near the service.

2. The experiments mainly show communication efficiency, not a clearly new capability. The workflows are constructed and fairly narrow, such as getting N items, modifying N items, downloading N files, or uploading N files. These settings are fine for latency studies, but they do not strongly establish that tool programs are a broadly better interface for realistic agentic tasks. Overall, a pertinent problem discussed by the article is efficiency under repeated tool interaction, but the evidence is still mostly about batching and turn reduction.

3. Reliability remains a serious concern. The paper reports failures in about 20 percent of runs in program mode, mostly from Rust to Wasm compilation, and most of these failures cannot be repaired and fall back to stepwise execution. This weakens the case for the new interface as a robust default.

---

> ### Author Rebuttal · Authors · 2026-03-31
>
> Dear Reviewer 3N7o,
>
> Thank you for your valuable feedback and insightful review. We will now address your concerns in detail. We hope our responses will help and ready to provide any further clarification.
>
> - **W1 & Q1. What is the precise novelty claim?**
>
> TOOLPRO does not merely use code or batch calls; it makes the effect-typed executable tool program itself the action representation. In web-service settings, plan-and-act/code-agent reasoning is still typically realized as individual tool calls at execution time from client side. TOOLPRO instead emits the side-effectful tool program itself as the action object, changing decision granularity, replay semantics, and consolidation behavior. More details see `hgUu`, W1/Q3.
>
> - **W2, Q2 & Q3. Evidence beyond narrow workflows, mixed-task fallback cost, and realistic WRITE semantics.**
>
> We have added the following realistic benchmarks.
>
> | Extended bench | Core challenge | Representative task/result |
> | --- | --- | --- |
> | `bench1` | Nondeterministic task | Find the customer Mei Patel (ZIP 76165), identify the correct pending order at runtime, and update that order to contain only item `1096508426` with total `55.0`.  |
> | `bench2` | Loop + branching + coordinated writes |  Update all pending orders and the customer’s primary address, and if order `#W4082615` has not shipped, replace its items with a simpler puzzle and set the total to `25.0`. |
> | `bench3` | Non-idempotent | For delivered order `#D12345`, change the status to `exchange_requested`, replace its items, and create a corresponding exchange log entry recording both old and new items.  |
> | `bench4` | Cross-service | Upload 5 local documents to MinIO, inspect each object’s text content, route it across storage prefixes, create follow-up memos when required, and then clean up all generated files and memos. |
>
> These benches go beyond simple batching or fixed loops: they require nondeterministic retrieval, control flow, intermediate binding, coordinated multi-record writes, non-idempotent side effects, and cross-service branching. Across a mixed measurement of bench1-3, TOOLPRO reduces latency from 30.16s to 17.91s (40.6%), improves task accuracy from 0.60 to 0.93 (gains arise from reduced intermediate decision errors and improved state consistency), and lowers client-side llm latency from 14.98s to 7.08s (-52.8%).
>
> In the mixed-task evaluation, all three benches remain net positive. Specifically, including cold-start runs, the policy selects program mode in 12/15 runs, with no fallback once program mode is selected, and matches the empirically better mode in 12/15 runs. After profiles are available (warm start), it selects program mode in 12/12 runs, again with no observed fallback, and matches the empirically better mode in all 12/12 runs.
>
> These gains come from reducing decision complexity of agent, not just communication overhead: many local next-call decisions are replaced by one executable program with control flow, reducing repeated client-side deliberation and intermediate-memory accumulation.
>
> On the separately reported bench4, TOOLPRO reduces latency from 52.68s to 24.54s (53.4%), cuts client-side traffic by 96.1%, and improves accuracy from 0.2 to 0.8. Bench4 adds runtime content inspection, cross-service branching, and side-effects. These results show that TOOLPRO remains effective under more realistic service behavior.
>
> - **W3. Program-mode failures weaken robustness.**
>
> To clarify reliability, we added (i) a model-comparison experiment on Rust-based bench2, and (ii) a breakdown of failures on bench2.
>
> | Model| Success rate | Note |
> | --- | --- | --- |
> | qwen3-coder-flash | 0.8 | Weaker code model |
> | gpt-5.1 | 1 | No compilation failure / no fallback observed |
> | gemini-3-flash-preview | 1 | No compilation failure / no fallback observed |
>
> Based on (i), the observed program-mode failures are primarily a model-quality bottleneck rather than an inherent limitation of TOOLPRO: on the same Rust-based benchmark, stronger code models reach 1.0 success rate with no compilation failure and no fallback observed. The previously reported 20% program-mode failure rate corresponds to earlier runs with weaker code models (qwen3-max).
>
> Based on (ii), the dominant failure is not unrecoverable compilation or repair failure. With replay enabled, we observe 0 compile failures, 0 repair failures, 0 replay-invalidating repairs, and 0 fallback across 5 program-mode attempts. In the matched no-replay ablation, the only failure-mode event is 1/5 conservative fallback to stepwise due to an unsafe write condition, and that run still succeeds.
>
> Additionally, TOOLPRO is fail-closed: when replay soundness or program construction cannot be guaranteed, the system falls back to the safer stepwise mode rather than risking incorrect side effects. Combined with the mixed-task results above, this shows that TOOLPRO ensure execution safety without causing much overhead, and that fallback does not reverse the overall advantage.

---

### Decision · Program_Chairs · 2026-04-30

**Decision:**

Accept (regular)

**Comment:**

ToolPro represents agent tool intent as executable tool programs with explicit effect types, enabling constraint-guided program construction, effect-aware replay for exactly-once WRITE semantics, and a profile-driven consolidation policy. Evaluated on three applications and six workflows, it reduces latency by up to 37.2% and client-side traffic by up to 85.3%.

Two reviewers were positive, finding the problem well-motivated and the system design clean. One reviewer maintained weak reject, citing proximity to existing code-based agent work (CodeAct, AFlow) and narrow benchmark scope, but did not submit a rebuttal acknowledgement. The rebuttal added more complex benchmarks (nondeterministic retrieval, coordinated writes, cross-service workflows), a replay ablation confirming safety benefits, a language comparison, and a detailed differentiation from related work — fully resolving concerns for two of three reviewers.